# Application of Real-Time PCR for the Detection and Quantification of Oomycetes in Ornamental Nursery Stock

**DOI:** 10.3390/jof7020087

**Published:** 2021-01-27

**Authors:** Alexandra Puertolas, Peter J. M. Bonants, Eric Boa, Steve Woodward

**Affiliations:** 1ANSES, Laboratoire de la Santé des Végétaux-Unité de Mycologie, Domaine de Pixérécourt-Bât. E, CS 40009, F54220 Malzéville, France; alexandra.puertolas@anses.fr; 2Department of Plant and Soil Science, School of Biological Sciences, University of Aberdeen, Aberdeen, Scotland AB24 3UU, UK; e.boa@abdn.ac.uk; 3Wageningen Plant Research, Business Unit Biointeractions & Plant Health, Wageningen UR, 6700 AA Wageningen, The Netherlands; peter.bonants@wur.nl

**Keywords:** TaqMan PCR, *Phytophthora*, *Pythium*, quantification, ornamental plants, international trade

## Abstract

Numerous *Phytophthora* and *Pythium* disease outbreaks have occurred in Europe following inadvertent introduction of contaminated ornamental plants. Detection and identification of pathogens are crucial to reduce risks and improve plant biosecurity in Europe and globally. Oomycete diversity present in roots and compost was determined in 99 hardy woody plants bought from nurseries, retailers and internet sellers, using both isolations and molecular analyses. Oomycete DNA was quantified using real-time PCR of environmental DNA from the plants using three loci: ITS, trnM-trnP-trnM and atp9-nad9. At least one oomycete species was isolated from 89.9% of plants using classical techniques. In total, 10 *Phytophthora* spp., 17 *Pythium* spp. and 5 *Phytopythium* spp. were isolated. Oomycetes were isolated from 86% of asymptomatic plants, but real-time PCR demonstrated that oomycetes were associated with all plants tested. More oomycete DNA occurred in composts in comparison with roots and filters from baiting water (a mean of 7.91 ng g^−1^, 6.55 × 10^−1^ ng g^−1^ and 5.62 × 10^−1^ ng g^−1^ of oomycete DNA detected in compost with ITS, trnM and atp9 probes, respectively); the ITS probe detected the highest quantities of oomycete DNA. No significant differences were found in quantities of oomycete DNA detected using real-time PCR in plants purchased online or from traditional retailers.

## 1. Introduction

In the last three decades, the global horticultural industry has grown exponentially due to the development of new technologies and improved packaging and shipping techniques, which have transformed international trade in ornamental plants [1]. Nurseries have improved propagation technologies to produce higher volumes of ornamental plants to satisfy consumer demands and the desire for ready-made gardens [2,3]. During expanded internationalization of trade over the last 50 to 60 years, many plant pathogens have been transported from their native geographical regions and introduced to other regions, leading to new and sometimes highly destructive disease outbreaks, damaging forest and riparian ecosystems and resulting in irreversible economic, social and biological losses [3,4]. Commerce in general, especially internet trade, has increased in importance over the last 20 years within different industrial sectors, including trade in ornamental plants. Various websites provide interactive platforms between small businesses and customers on an international scale. Plants purchased through internet sites are posted to customers and might carry pathogens that pose threats to destination states as the suppliers, especially in countries less well-regulated than Europe, may not follow legal requirements imposed by National Plant Protection Organizations (NPPOs). Plants from these sources entering different territories are less likely to be inspected and, therefore, pose a high risk [4,5,6,7,8].

The genera *Phytophthora* and *Pythium* include many damaging plant pathogens, which affect agricultural and horticultural crops and forest ecosystems [3,9]. Increasing evidence suggests that the main dispersal pathway for oomycete plant pathogens is through the international nursery trade, particularly on potted plants that include soil substrates or compost [1,2,4,8,9,10,11,12]. The spread of *Phytophthora ramorum* across Europe through trade in ornamental plants has been well documented [10,13], despite the pathogen being listed as a major quarantine species on the EPPO (European and Mediterranean Plant Protection Organization) A2 list. With the intensification of nursery surveys due to *P. ramorum* disease outbreaks, at least sixteen previously unrecognised *Phytophthora* species were described affecting ornamentals since 2000; moreover, since 1980, over 30 *Phytophthora* species have been isolated, described and reported on woody ornamentals around the world [2].

Compared to classical techniques, molecular methods have greatly reduced the time and costs for identifying oomycete pathogens. Such methods are highly sensitive and can detect microorganisms present in low abundance. Molecular detection and quantification using real-time PCR approaches have been applied to detect oomycetes in soil and plant tissues [14,15]. Real-time PCR improves the speed, sensitivity and accuracy of DNA amplification in comparison with standard PCR [14,15,16], and has been used to identify and quantify *Pythium* and *Phytophthora* species including *Phytophthora ramorum* and *P. kernoviae* [17,18,19,20], which cause diseases of natural vegetation and horticultural crops. Different loci have been used to develop real-time PCR assays using TaqMan chemistry [21], the ITS region being most commonly selected for the design of primers and probes [14,17]; other loci have also been used, however, including mitochondrial loci such as atp9-nad9 and trnM-trnP-trnM [21]. DNA can be extracted from environmental samples, such as soil, water and air (environmental DNA or eDNA) [22], prior to isolating a target organism. eDNA represents the mixture of organisms present in a sample, though degradation may occur. Analysis of eDNA using metabarcoding is a rapid and cost-effective technique for assessing the diversity of oomycetes present, compared to cloning and Sanger sequencing [22].

The aim of the work described in this paper was to apply a combination of classical and molecular methods to study the diversity of oomycetes present on hardy ornamental nursery stock at the point of sale, and to quantify the pathogen load in roots, plant compost and the water from baiting, using three different loci. The hypotheses tested were that (1) asymptomatic plants carry high loads of oomycete DNA; and (2) plants purchased through internet sales present a risk of international plant pathogen dissemination without necessarily passing through routine inspections at state borders. Pathogen load was quantified in plants with or without aerial symptoms of infection, and comparisons of oomycete loads made between plants obtained from different sources (direct purchase from nurseries and other outlets, or from internet sales).

## 2. Materials and Methods

### 2.1. Plant Samples and Oomycete Isolation and Identification

Ninety nine woody ornamental plants, including 23 species of ornamentals and/or cultivars commonly imported into Europe, were analysed. Plants were bought from retail outlets, including supermarkets, and various nurseries in the United Kingdom and the Netherlands, and through internet purchases, via the Amazon and eBay websites. In total, 49 plants were obtained in the UK and 50 in the Netherlands. Plants were chosen randomly when bought in physical shops, without paying attention to visible symptoms. Plants bought in the Netherlands were shipped by overnight courier to University of Aberdeen, where all isolations and baiting tests were carried out.

Plants were inspected visually and any visible symptoms on above-ground plant parts photographed. Aerial plant parts were cut off and discarded, and the root systems carefully separated from the compost. Roots were washed under running tap water and flooded with distilled water to remove debris and enhance oomycete activity. After 3–4 h, visible lesions on the roots were plated directly onto CMA-P_5_ARBP/H selective medium [23] in 90 mm diam. Petri dishes and incubated at 25 °C in the dark, with inspection at 24 h intervals. Roots were air-dried in paper envelopes at room temperature until required for direct extraction of DNA.

Plant composts were analysed by baiting with apple and leaf assays [24]. Granny Smith™ apples were surface sterilised by wiping with tissue paper soaked in 100% ethanol. Four equidistant holes were made perpendicularly around each apple using a 15 mm diam. cork borer. Each hole was filled with 5–10 g of compost, flooded with sterile distilled water and sealed with tape to avoid desiccation. Apples were incubated at room temperature in ambient conditions. When lesions were visible on the apple surfaces, tissues from the active margins of the internal lesions were plated directly onto CMA-P_5_ARBP/H to obtain pure isolates.

For the leaf baiting assay, potting composts were placed in plastic containers (20 × 20 × 7 cm^3^) up to 3 to 4 cm deep and saturated with sterile distilled water. Plastic boxes were transferred to a glasshouse for 24 h. More distilled water was then added to immerse the compost completely and three to four leaves of *Rhododendron* spp. (*R. concinnum*, *R. decorum*, *R. agustinii, R. fortuneii*) floated on the water surface. Baiting experiments were maintained at 25 °C under daylight conditions in the greenhouse. When lesions appeared on the foliage, leaves were washed in distilled water, patted dry on tissue paper and active margins cut from the lesions and plated onto CMA-P_5_ARBP/H. Cultures were incubated at 25 °C in the dark. Compost from the leaf baiting assays were recovered and oven dried at 30 °C for 7 days in preparation for DNA extraction.

Hyphal growths appearing on the CMA-P_5_ARBP/H were examined under a binocular microscope and hyphal tips transferred to potato dextrose agar (39 g L^−1^, PDA, Oxoid, Thermo Fisher Scientific, Waltham, MI, USA) to obtain pure cultures. Cultures on PDA obtained from roots, apples and baiting assays were classified by colony morphology and identified by DNA sequencing. DNA extractions from pure cultures were made using a previously published protocol [25] and identification was based on amplification of the Internal Transcribed Spacer region (ITS1-5.8S-ITS2) of ribosomal DNA by PCR using ITS4 and ITS6 primers [26,27]. Amplified samples were purified with the EZNA Cycle Pure Kit (Omega Bio-Tek, Norcross, GA, USA) and sequenced by Source Bioscience Lifesciences or Macrogen Europe. Sequencing results were analysed with CLC Main Workbench (Qiagen, San Diego, CA, USA) by comparison against two different databases to identify each isolate: GenBank, using the BLAST tool with the algorithm “blastn”; if the result suggested a *Phytophthora* species, the sequence was compared against accessions in the *Phytophthora* Database (http://www.phytophthoradb.org/).

### 2.2. Extraction of Environmental DNA (eDNA)

DNA extractions were made from roots, compost and filters used to sample water from the baiting assays (see below) from each nursery plant, using a modification of the method of Català et al. [28,29]. eDNA was used to quantify the pathogen load in each plant source by TaqMan PCR. DNA was extracted from three sub-samples of compost, roots and filters for each plant; subsequently, 20 µL aliquots of each DNA extraction was mixed for use in the TaqMan PCR. A negative control of DNA extraction was performed using clean filters and without plant compost or roots, and negative amplification of the target organisms was assessed using standard PCR with ITS4 and ITS6 primers. All DNA extractions were kept frozen at −20 °C until PCR to avoid degradation.

Water (approx. 180 mL) from each baiting experiment was passed through a nylon mesh (36 µm) to remove compost particles before passing through 45 mm diam., 5 µm pore size nitrocellulose Millipore filter membranes (Merck Millipore, Darmstadt, Germany). Filters were stored at −20 °C until eDNA extraction. Filters were crushed and homogenised in liquid nitrogen and eDNA extracted from 100 mg filter powder using the PowerSoil DNA Isolation kit (MoBio Laboratories, Carlsbad, CA, USA), following the manufacturer’s protocol. eDNA was eluted in 100 µL TE buffer (10 mM tris, 1 mM EDTA, pH 8.0, MoBio Laboratories, Carlsbad, CA, USA).

Roots were air-dried in paper envelopes at room temperature, homogenised in liquid nitrogen and stored at −20 °C until DNA extraction. DNA was extracted from 50 mg root powder using the PowerPlant Pro DNA Isolation kit (MoBio Laboratories, Carlsbad, CA, USA), following the manufacturer’s protocol, including addition of the phenolic removal solution. eDNA was eluted in 100 µL TE buffer.

Plant compost was recovered from the baiting experiment by draining the water, before drying the composts in an oven at 30–40 °C for one week. Composts were sieved (1.18 mm mesh size) to remove larger particles and other debris, and ground in a Retsch PM100 Ball Mill (Retsch, Haan, Germany) at 500 rpm for 20 s. Substrate samples were maintained at 4 °C until DNA extraction. DNA was extracted using the PowerSoil DNA Isolation kit following the manufacturer’s protocol, but modifying the lysis step [30], by adding 1.2 mL saturated phosphate buffer (Na_2_HPO_4_; 0.12 M, pH 8) to 100 mg of pulverised compost sample for the recovery of extracellular DNA [30,31]. The mix was homogenised using a vortex at maximum speed for 15 min and centrifuged at 20,000× *g* for 10 min before continuing with the DNA extraction protocol. eDNA was eluted in 100 µL TE buffer.

### 2.3. Detection and Quantification of Pathogen Load in eDNA Extractions from Nursery Plants Using Real-Time PCR

Real-time PCR was used to detect and quantify the oomycete load in environmental DNA (eDNA) extractions from filters, roots and composts. TaqMan assays were performed using three different loci (Table 1): Internal Transcribed Spacer (ITS) region (All_Phy_probe and FITS_15Ph, RITS_279Ph primers designed by Kox et al. [32]; trnM-trnP-trnM region (using TrnM_PhyG_probe2 and primers PhyG-F2, PhyG-Rb) and atp9-nad9 region (ATP9_PhyG2_probeR and primers PhyG_ATP9_2FTail, PhyG_R6_Tail) [21].

The standard curve used for the quantification of eDNA was constructed for each experiment using genomic DNA extracted from *Phytophthora ramorum* (isolate 589, PRI collection, Wageningen University), previously identified and quantified with Quan-iT PicoGreen (Applied Biosystems, Foster City, MI, USA). Ten-fold serial dilutions were made from DNA of *P. ramorum* from 1 to 10^−5^ ng µL^−1^ and two replicates of each dilution included with each reaction to create a standard curve for interpolation of target results and to obtain eDNA quantifications. Reaction efficiencies were calculated automatically with the 7500 Software v2.3 (Applied Biosystems, Foster City, MI, USA).

TaqMan PCR assays were carried out in an ABI 7500 Real-Time PCR System (Applied Biosystems, Thermo Fisher Scientific, Waltham, MI, USA) on white 96 well plates. The amplification mix with the IPC and ITS probes contained 1x Takara Premix Ex Taq Perfect Real Time, including 1.25 U/25 µL of Takara Ex Taq HS, dNTPs each at 0.4 mM and 4 mM Mg_2_^+^ (Takara, ClonTech Laboratories, Japan), 0.5x ROX Dye II as passive reference, 0.3 µM each amplification primer and 0.1 µM each TaqMan probe, 1 µL undiluted eDNA, 0.6 µM each internal control primer, 0.13 µM internal control probe, 0.1 ng µL^−1^ plasmid PLRV DNA and Milli-Q water to a final reaction volume of 25 µL. For the trnM and atp9 probes the amplification mixture was the same but the IPC primers, probe and plasmid DNA were excluded and 2 µL of undiluted eDNA added. Negative controls were included using Milli-Q water as template in all PCRs for all plant composts (ITS, trnM and atp9 with the plasmid PLRV DNA). Cycling conditions were 95 °C for 2 min; 50 cycles of 95 °C at each specific annealing temperature for 1 min. Data were collected in the last holding stage of each cycle. Results were analysed using 7500 Software v2.3 (Applied Biosystems, Thermo Fisher Scientific, Waltham, MI, USA) and exported to an Excel template to determine eDNA concentrations. eDNA concentrations were determined by absolute quantification, extrapolating the eDNA concentration using Ct (cycle threshold) values obtained on each measure from the logarithmic regression line of each standard curve generated.

All TaqMan probes used in this work were tested to check for genera specificity using DNA extracted from pure cultures of several *Phytophthora* (*P.* x *cambivora*, *P. cinnamomi*, *P. cryptogea*, *P. nicotianae* and *P. ramorum*), *Pythium* (*P. dissotocum*, *P. irregulare*, *P. rostratifingens*, *P. sylvaticum* and *P. undulatum*) and *Phytopythium* species (*P. chamaehyphon*, *P. helicoides*, *P. litorale* and *P. vexans*). The trnM and atp9 probes were specific to *Phytophthora* species and did not amplify *Pythium* or *Phytopythium* spp. However, the ITS probe amplified all *Phytophthora* species tested and some of the *Pythium* and *Phytopythium* spp.

### 2.4. Internal Positive Control (IPC) to Detect Inhibitors in eDNA

To detect the potential presence of inhibitory substances, such as humic and fulvic acids, an internal positive control (IPC) designed by Waalwijk et al. [33] was included in the TaqMan PCR reactions in duplex with the ITS probe [15,34]. All target probes were labelled at the 5′ end with the fluorescent reporter dye 6-FAM (6-carboxyfluorescein) and the IPC probe with HEX, while the 3′ end was modified with a Black Hole Quencher-1 (BHQ-1) in all probes (Table 1). The possible presence of inhibitory substances was assessed by comparison of Ct values from the negative control of the oomycete TaqMan probes with the Ct value of the IPC probe in the sample. Samples showing inhibition of TaqMan amplification were diluted 1:10 and TaqMan PCR assays repeated.

The IPC assay was previously optimized to determine the optimal IPC plasmid DNA concentration to perform target amplification without reaction inhibition due to high amounts of IPC DNA that could consume all PCR reagents. A serial dilution of IPC DNA was carried out from 10^−3^ to 10^−6^ ng µL^−1^. Subsequently, a dual-TaqMan reaction was performed with the three TaqMan probes with *P. ramorum* gDNA at the lowest concentration amplified with these probes without the IPC (10^−4^ ng µL^−1^ for the ITS and trnM probes and 10^−3^ ng µL^−1^ for the atp9 probe). Ct values of the IPC, the negative control and *P. ramorum* gDNA were compared to assess the detection limit of the probes. The optimal concentration of the IPC selected was 10^−4^ ng µL^−1^ for all probes, where the Ct value of the IPC had a comparable value to the Ct of the negative control.

Differences between the quantification results using each of the three probes and the types of samples (filters, roots, composts) were examined using one-way ANOVA applied on log transformed data, and Tukey’s HSD post hoc tests were carried out (*p* = 0.05). To compare DNA quantities within symptomatic or asymptomatic plants, and to determine differences according to the sale source (online or retailer), *t*-tests were performed on log transformed data. All statistical analyses were performed using R software (v. 3.5.0) [35].

## 3. Results

### 3.1. Isolation and Identification of Oomycetes

Ninety-nine hardy ornamental nursery plants were analysed, of which 67 were purchased online in the UK and the Netherlands (Table 2). The plants presented a wide range of symptoms, from healthy, to withered foliage, chlorosis, necrosis on leaves and roots, to defoliation and collapse. Information on country of origin was provided for only twelve plants: six plants (P2 to P7, *Acer palmatum*) were from China and another six (P14 to P19, *Rhododendron* spp.) from Germany. Approximately 35% of plants were symptomless (35 plants out of 99) but 86% of these symptomless plants were infected with, or had at least one species of *Phytophthora*, *Pythium* or *Phytopythium* present in the compost (Table 2).

Oomycete species from the three genera targeted were isolated from 89.9% of all plants sampled. *Phytophthora* spp. were detected in 41.4% of the plants analysed, *Pythium* spp. in 76.8% of plants and *Phytopythium* spp. in 46.5% (Figure 1). In total, 10 *Phytophthora* spp., 17 *Pythium* spp. and 5 *Phytopythium* spp. were isolated using classical techniques, from the 23 plant species and cultivars analysed (Figure 2, Table 2).

The *Phytophthora* species isolated most frequently was *P. cryptogea*, from 13.1% of plants, followed by *P.* x *cambivora* and *P. citrophthora*, both isolated from 10.1% of plants (Figure 2). Of the *Pythium* species, *P. dissotocum* was isolated from 54.5% of the plants tested, followed by *P. undulatum* in 9.1%. The most common *Phytopythium* isolated was *P. litorale*, present on 34.3% of the plants, followed by *P. chamaehyphon* (10.1%) (Figure 2, Appendix A).

### 3.2. Quantification of Oomycota with TaqMan PCR

Quantification assays using TaqMan probes successfully detected and quantified oomycete species in eDNA samples. Standard curves produced in each assay showed correlation coefficient values (R^2^) of between 0.98 and 0.99. Efficiencies [E = (10(−1/slope) − 1) × 100] were 77–86% in the ITS assays, 65–76% in trnM assays and 75–89% in atp9 assays. Amplifications with a Ct value higher than the Ct value obtained on the last dilution of the genomic DNA of the standard curve were assessed manually and not included in the analysis if shown to be negative.

TaqMan PCR assays using the ITS probe detected oomycete DNA in 90.9%, 97% and 90.9% of the filters, roots and composts sampled, respectively. With the *Phytophthora* specific probe trnM, *Phytophthora* DNA was amplified in 27.3%, 24.2% and 40.4% and with the atp9 probe 36.4%, 41.4% and 57.6% of the analysed filters, roots and compost samples, respectively (Figure 3). The oomycete general ITS probe obtained on average higher amounts of oomycete eDNA in all types of samples (filters, roots and plant composts), in comparison with the *Phytophthora* specific probes (Figure 4). This ITS probe detected 2.83 × 10^−1^ ng g^−1^ of oomycete eDNA on filter samples, 4.28 × 10^−1^ ng g^−1^ on root samples and 7.91 ng g^−1^ on plant composts, with significant differences between the three types of Samples analysed (ANOVA, F (2, 271) = 54.06, *p* < 0.001).

The *Phytophthora* specific trnM probe detected 81.98 ng g^−1^
*Phytophthora* DNA on filters (due to a high amount of DNA detected in sample P99), 2.48 × 10^−2^ ng g^−1^ on roots and 6.55 10^−1^ ng g^−1^ on plant composts. Differences were found within the type of sample analysed (ANOVA, F (2, 88) = 9.48, *p* < 0.001), but not between root and filter samples (Tukey HSD post hoc test, *p* = 0.62). The atp9 probe, also *Phytophthora* specific, detected 2.49 × 10^−1^ ng g^−1^ on filter samples, 1.96 × 10^−1^ in root samples and 5.62 × 10^−1^ ng g^−1^ in plant composts. However, using this probe, no significant differences were found between the type of sample analysed (ANOVA, F (2, 131) = 0.68, *p* = 0.51). Significantly greater quantities of DNA were detected using the oomycete general ITS probe (pooling all types of samples tested; filters, roots and composts) in comparison with the *Phytophthora* specific trnM and atp9 probes (ANOVA, F (2, 496) = 28.13, *p* < 0.001). Differences between the quantities of DNA detected with the *Phytophthora* specific trnM and atp9 probes were not significant (Tukey HSD, *p* = 0.25).

The highest oomycete eDNA concentration obtained in plant composts was found with the ITS probe in the compost of plant P22 (*Acer palmatum* ‘Dissectum filigree’) with 191.34 ng g^−1^ compost, followed by the trnM probe with 9.91 ng g^−1^ and the atp9 probe with 6.59 ng g^−1^, both from plant sample P14 (*Rhododendron* ‘Germania’) (Appendix A). On filter samples, the highest amounts of eDNA were detected with the *Phytophthora* specific trnM probe, from plant P99 (*Ceanothus* ‘Southmead’) with 2212.20 ng g^−1^ filter. The atp9 *Phytophthora* specific probes detected a maximum amount of eDNA of 6.03 ng g^−1^, also from plant P99, meanwhile maximum detection of eDNA with the general probe was on filters from plant P8 (*Hebe* x *franciscana* ‘Variegata’) with 4.17 ng g^−1^. On root samples, the highest concentrations of eDNA amplified were detected with the ITS probe, with 10.94 ng g^−1^ from roots of plant P61 (*Ilex meserveae*), followed by 2.28 ng g^−1^ in roots of plant P1 (*Viburnum* x *bodnantense*) using the *Phytophthora* atp9 probe, and 9.94 × 10^−2^ ng g^−1^ of roots on plant P75 (*Pinus mugo*) using the trnM probe (Appendix A). The lowest quantity of eDNA detected using the TaqMan assay on average, was found on root samples, with 6.76 × 10^−4^ ng g^−1^ using the ITS probe, 1.70 × 10^−4^ ng g^−1^ and 1.21 × 10^−4^ ng g^−1^ with the *Phytophthora* specific trnM and atp9 probes, respectively (Appendix A).

Oomycete eDNA was detected in all asymptomatic plants with at least one of the tested probes: with the ITS probe, roots of 100% of asymptomatic plants, 91.4% of filters and 85.7% of composts were positive. The trnM probe detected *Phytophthora* eDNA in the compost of 45.7% of asymptomatic plants, 37.1% of roots and filters, whereas the atp9 probe detected eDNA in 54.3% of asymptomatic plant composts, 48.6% of roots and 37.1% of filters (Figure 5). No significant differences were found between the means of DNA quantified in symptomatic and asymptomatic plants, with either the general oomycete ITS probe or the *Phytophthora* specific probe trnM (Figure 6): ITS probe *t*-test t (202.73) = 1.44, *p* = 0.15 (M = 1.59 × 10^−1^ ng g^−1^, SE = 1.19 in symptomatic plants and M = 2.43 × 10^−1^ ng g^−1^, SE = 1.26 in asymptomatic plants); trnM probe *t*-test t (88.92) = 0.37, *p* = 0.71 (M = 2.66 × 10^−1^ ng g^−1^, SE = 1.47 in symptomatic plants and M = 3.24 × 10^−2^ ng g^−1^, SE = 1.44 in asymptomatic plants). However, differences were found within symptomatic and asymptomatic plants using the *Phytophthora* specific atp9 probe (Figure 6): *t*-test t (114.16) = 4.35, *p* < 0.001 (M = 2.75 × 10^−2^ ng g^−1^, SE = 1.27 in symptomatic plants and M = 1.30 × 10^−1^ ng g^−1^, SE = 1.30 in asymptomatic plants). The greatest amount of oomycete DNA detected on asymptomatic plants, 47.81 ng g^−1^ of compost, was found in the compost of plant P15 (*Rhododendron* ‘Germania’) using the general ITS probe followed by the *Phytophthora* specific trnM and atp9 probes on plant P14 compost (*Rhododendron* ‘Germania’), with 9.91 ng g^−1^ and 6.59 ng g^−1^, respectively (Appendix A). *Phytophthora cinnamomi* was consistently isolated from these plants using baiting and direct plating methods. Moreover, other *Phytophthora* species were isolated from asymptomatic plants, including P50 to P54, P64 and P73 from which *P. nicotianae*, *P. cactorum*, *P. cinnamomi, P. chlamydospora, P.* x *cambivora* and *P. citrophthora* were obtained; these species were also detected with the *Phytophthora* specific probes atp9 and trnM.

There were no significant differences between the amount of DNA detected on plants bought through internet platforms and those obtained directly from nursery retailers and shops with any ofthe TaqMan probes tested (Figure 7): ITS probe *t*-test, t (240.48) = −0.98, *p* = 0.33 (M = 1.58 × 10^−1^ ng g^−1^, SE = 1.26 for plants obtained on the internet and M = 2.10 × 10^−1^ ng g^−1^, SE = 1.20 for plants obtained in nurseries); trnM probe *t*-test, t (79.24) = 1.13, *p* = 0.26 (M = 4.11 10^−2^ ng g^−1^, SE = 1.53 for internet purchases and M = 2.23 × 10^−1^ ng g^−1^, SE = 1.40 for nursery retailer); and atp9 probe *t*-test, t (107.98) = 0.02, *p* = 0.98 (M = 4.88 10^−2^ ng g^−1^, SE = 1.36 for internet purchases and M = 4.83 × 10^−2^ ng g^−1^, SE = 1.28 for plants obtained in nurseries).

## 4. Discussion

Using real-time PCR with three TaqMan probes suggested that a high number of oomycete species were associated with the woody ornamental plants tested, particularly in the plant composts. More oomycete DNA was detected using the ITS probe due to a general specificity for the three genera studied, *Phytophthora*, *Pythium* and *Phytopythium,* whereas the trnM and atp9, being *Phytophthora*-specific, detected less DNA, with no differences in detection rates between these two probes. Not all *Phytophthora* and *Pythium* species are culturable and detectable using baiting, whereas DNA based techniques, such as real-time PCR, detect not only living organisms, but also DNA from dead or moribund organisms. Some of the plants tested were heavily infected by *Phytophthora* species. In addition, *Phytophthora* spp. were also detected on asymptomatic plants such as P50, a *Ceanothus thyrsiflorus* ‘Repens’, from which *P. nicotianae* was consistently isolated through compost baiting. Oomycete DNA was also detected in the compost of this plant with the general ITS probe. Results of the real-time PCR assays with the oomycetes general probe and *Phytophthora* specific probes correlated with the species of *Phytophthora* and *Pythium* obtained by isolation methods on each plant. The detection of oomycetes only using real-time PCR could indicate the presence of oomycete propagules in very low quantities, which would be difficult to isolate using traditional methods, or the presence of unculturable or non-viable oomycete species.

Copy numbers of each locus may vary in different genera and within species [36]. These variations were previously reported for rDNA of true fungi [36,37,38]. Copy numbers for oomycete species have yet to be determined, but it is presumed that variations also occur in these organisms [21,36]; for example, it is estimated that there are approximately 100 to 150 copies of ITS and some 60 copies of mitochondrial DNA per genome.

Water from compost baitings was analysed to quantify live inoculum present in compost. The water from the baiting tests was collected carefully, recovering only the supernatant, which should contain active zoospores, but is less likely to contain mycelial fragments compared to the compost itself. Therefore, lower quantities of oomycete DNA were found using TaqMan probes on eDNA from filter samples than from roots or plant composts (Appendix A). More oomycete DNA was detected in plant composts than in roots, implying that not all the oomycetes detected were infecting the plants.

High numbers of oomycete infections were also found in asymptomatic ornamental plants by Migliorini et al. [12] using real-time PCR. Approximately 70% of asymptomatic plants were contaminated by one or more species of *Phytophthora*, *Pythium* and *Phytopythium*. In the work described here, however, the ITS probe detected oomycetes in the roots of all asymptomatic plants, along with 91.43% of filters and 85.71% of composts. Asymptomatic plants such as P9 (*Hebe* x *franciscana* ‘Variegata’) included large quantities of oomycete DNA in the composts (14.19 ng g^−1^ of DNA using the general oomycete ITS probe), or P77 (*Pinus mugo*) compost, containing 3.43 ng g^−1^ and 5.74 ng g^−1^ of *Phytophthora* DNA detected with trnM and atp9 probes, respectively. These results confirmed the high rates of oomycete species present in ornamental plants reported by Migliorini et al. [12]. Other studies carried out by Prigigallo et al. [39,40] demonstrated the wide diversity of *Phytophthora* species present in ornamental plants using molecular methods (semi-nested PCR and metabarcoding), including some putative novel species.

*Phytophthora ramorum* was amongst the 10 *Phytophthora* spp. isolated using classical techniques, from a dying *Viburnum* x *bodnantense* ‘Dawn’. *Phytophthora ramorum* has spread throughout Europe in the ornamental plant trade [13,41] and arguably poses a great risk to woody plant species. In the UK, *P. ramorum* is causing dieback and death of Japanese larch (*Larix kaempferi*), along with many species of Ericaceae and *Viburnum* [2,13].

Several *Phytophthora* species detected in this work have been described affecting and damaging forest ecosystems. For example, *P.* x *cambivora*, *P. cactorum* and *P. cinnamomi* are all involved in Ink Disease, which causes high mortality in forests and orchards of sweet chestnut (*Castanea sativa*) in Europe [42]. *Phytophthora cinnamomi* also causes decline and mortality of cork and holm oak (*Quercus suber*, *Q. ilex*) in southern Spain, Portugal and Italy [43,44,45]. In this study, *P. cinnamomi* was isolated in abundance from asymptomatic *Rhododendron* plants (e.g., P14 and P15). The results from the isolation work correlated positively with those from the TaqMan assays, as *P. cinnamomi* was also detected abundantly using both oomycete general and *Phytophthora*-specific probes in composts.

The most common *Pythium* species isolated in this work was *P. dissotocum*, which has been widely reported causing seedling damping-off and root disease in both agricultural crops and forestry plants. It is a virulent pathogen on Douglas-fir seedlings [46], on lettuce grown in hydroponics [47] and on ornamental plants [48]. *Pythium kashmirense,* first described by Paul & Bala [49] from Himalayan forest soils, was isolated in this work from compost baitings of *Viburnum plicatum* ‘Lanarth’ obtained from a nursery in the South-West of England. This paper is the first report of this species on ornamental plants and in the UK. Previously, *P. kashmirense* was found attacking soybean crops and in wetlands, both in the USA [49,50,51]. The *V. plicatum* ‘Lanarth’ plant was bought from an Internet retailer, highlighting the potential for long-distance dispersal of plant pathogens through this pathway.

In general, plant origins remain unknown to the end customer: plant retailers and nurseries rarely report the origin of a plant, or the propagation method used to raise plants, making monitoring and tracking of pathogen movement within and between different geographical areas difficult. Of the plants sampled here, only 12 of 99 included information on the country of origin (China and Germany). Such data would be of great value in the development of contingency measures following a disease outbreak once plants are planted [5,6]. Internet plant sales and rapid postal deliveries might also be speeding up the spread and establishment of potentially harmful pathogens, reducing the time plants spend in transit and delivering viable pathogens along with the plants. During this work, there were no differences in detection of oomycete species in plants obtained online and from physical retailers.

A major review published recently illustrated the high incidence of *Phytophthora* infestations in plant nurseries and the movement of these potentially damaging pathogens to gardens, forests and natural ecosystems [3]. The review emphasised the similar recovery rates of these pathogens in both containerised plants and bare root plants raised in field soils at nurseries, and on plants bought through internet platforms and nurseries. However, Jung et al. [3] also highlighted the fact that, despite the possible use of containerised plant production to control and manage *Phytophthora* infestations, poor nursery practices, the lack of simple tests to detect the presence of oomycetes and the complexity of the international plant distribution chain, all increase the risk of spread of *Phytophthora* species. Wholesale nurseries and plant retailers are intermediate steps between the original nursery where the plants were propagated and the end-customer. This trade network amplifies the probability of long distance spread of pathogens and escape to non-native ecosystems, increasing the probability of new disease outbreaks [1,2,3,4]. In 2010, over 4 billion plants for planting were imported into Europe from other countries [3,11], with the Netherlands as the main hub for imports. Different plant origins are registered as the source of imported plants, with Africa, followed by Asia and North and South America the main exporters. This huge number of plant imports greatly exceeds inspection capacity.

Some countries (Australia, New Zealand, USA) have highly restrictive regulations on importing plants and plant products to minimize the phytosanitary risk posed by pest and disease incursions. Europe follows EU Directive 2000/29/EC (amended by the Implementing Directive EU 2017/1279) on introduction of organisms harmful to plants, and the specific EU Regulation No 1143/2014 for the prevention of introduction of invasive alien species, but these regulations are less restrictive than those applied in North America, Australia or New Zealand, being based more on known pests/pathogens and paying more attention to preventive and contingency measures to avoid spread and establishment when the problem is first detected in the EU. Phytosanitary inspections based on visual controls of specific plant hosts are not sufficiently rigorous to detect the presence of many plant pathogens, partly due to the huge numbers of plants involved. Moreover, the focus on known pathogens markedly reduces the probability of detecting unknown plant pathogens [3].

To improve the detection capacity of phytosanitary inspections at borders, it is necessary to utilize accurate, reliable and rapid techniques. Current EPPO recommendations for screening for *Phytophthora* on ornamental plants include the use of ELISA immuno-detection assays, but these techniques give many false negative results [52]. Molecular detection methods based on DNA, also recommended as diagnostic tools by EPPO, are increasingly recognized as the most precise and reliable technologies available for application in this field, sensitive enough to detect target DNA in very low concentrations, whilst still providing identification to the species level [15,52]. The internal positive control (IPC) used in the TaqMan reactions in the present work gave consistent and reliable results when quantifying oomycetes, as reported previously for environmental samples [15,32,33]. In the current work, the IPC was used only for the ITS probe, as it greatly reduced the efficiencies of the trnM and atp9 probes. Reasons for low overall TaqMan reaction efficiencies could include the presence of inhibitors in the DNA extracts (which was controlled using the IPC), but also because reaction efficiency can vary when amplifying different parts of the genome [53]. The presence of secondary structures (like hairpins or loops) or high GC content of the target regions could also affect efficiency [54].

## 5. Conclusions

The two hypotheses tested in this work were accepted: (1) asymptomatic hardy woody plants do carry high loads of oomycete DNA; and (2) plants purchased through internet sales present a risk of plant pathogen dissemination internationally, although this particular risk is no greater than in conventional trade. This work represents a broad survey of ornamental plants in the international plant trade and applied the most up-to-date available molecular methods to detect oomycete movement on symptomatic and asymptomatic plants. The threat posed by oomycete pathogens transported internationally in the plant trade was confirmed [2,3,4,7,10,11]. Using three TaqMan probes, one general for oomycetes and two specific for *Phytophthora* species, enabled robust detection of oomycetes. The more general ITS probe was efficient in detection of oomycete DNA whereas the trnM and atp9 probes specifically detected *Phytophthora* species. These results clearly demonstrated the abundance of these plant pathogens being moved in the plants for planting pathway, not only in infected plants, but also in the compost.

## Figures and Tables

**Figure 1 jof-07-00087-f001:**
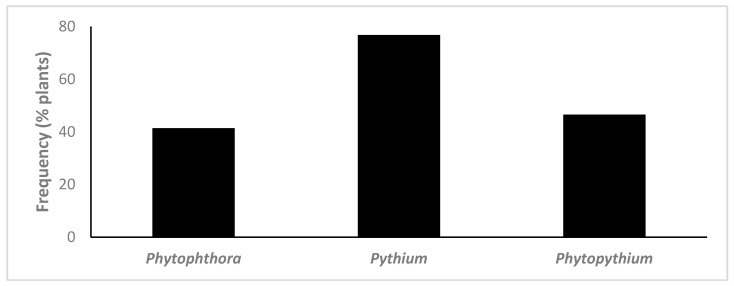
Frequency of detection of *Phytophthora*, *Pythium* and *Phytopythium* species in plants using classical isolation methods (*n* = 99).

**Figure 2 jof-07-00087-f002:**
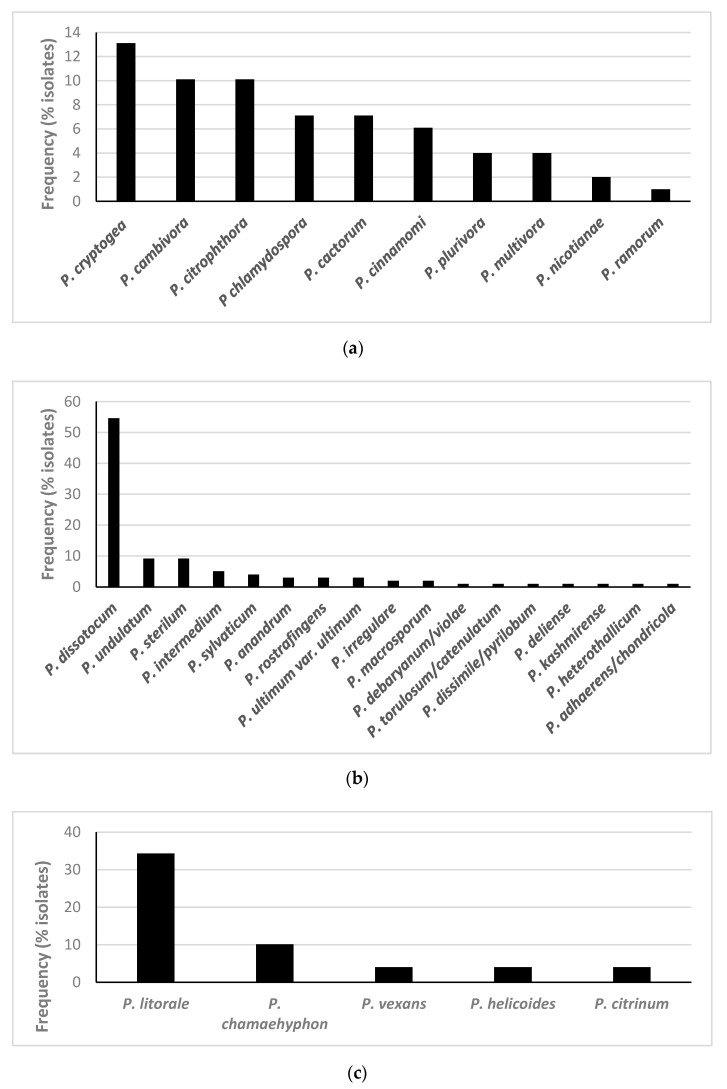
Frequency of isolation of species of (**a**) *Phytophthora*, (**b**) *Pythium* and (**c**) *Phytopythium* using classical methods from roots, composts and baitings from the 99 plants analysed.

**Figure 3 jof-07-00087-f003:**
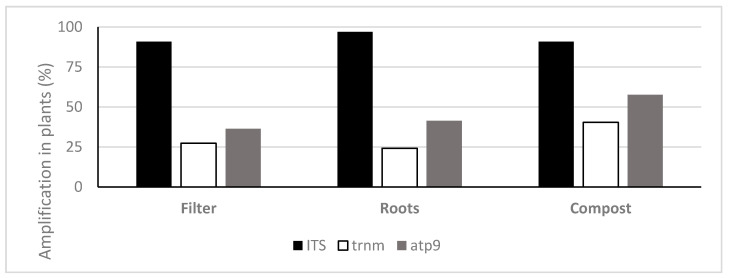
Frequency of detection of oomycete DNA using TaqMan probes on filters, roots and plant composts (*n* = 99).

**Figure 4 jof-07-00087-f004:**
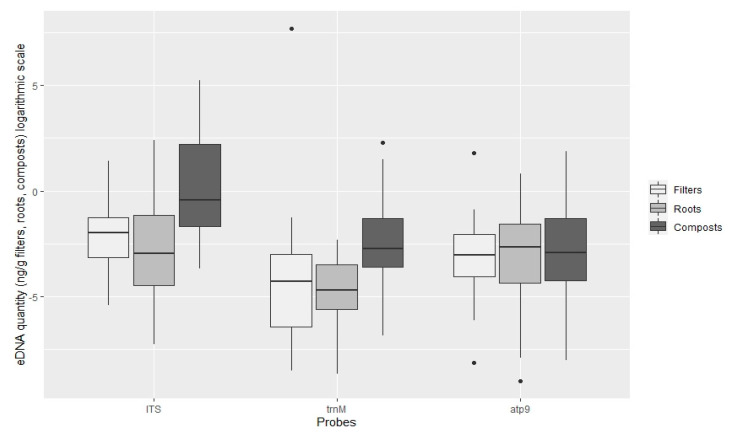
Quantification of oomycete eDNA (ng g^−1^, logarithmic scale) by TaqMan probe on filters, roots and plant composts. Boxplots indicate the 25th percentile (Q1, lower boundary), median (black line) and 75th percentile (Q3, upper boundary). Upper and lower whiskers indicate 10th and 90th percentiles. Dots above and below the whiskers represent outliers.

**Figure 5 jof-07-00087-f005:**
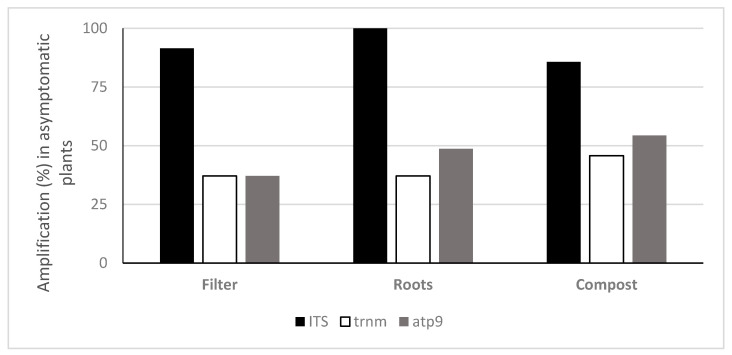
Frequency of detection of oomycete eDNA using TaqMan probes in asymptomatic plants.

**Figure 6 jof-07-00087-f006:**
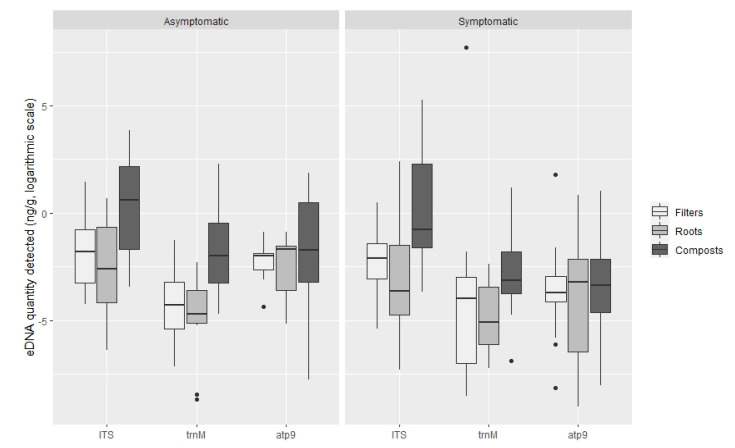
Quantification of oomycete eDNA (ng g^−1^, logarithmic scale) in symptomatic and asymptomatic plants by TaqMan probe. Boxplots indicate the 25th percentile (Q1, lower boundary), median (black line) and 75th percentile (Q3, upper boundary). Upper and lower whiskers indicate 10th and 90th percentiles. Dots above and below the whiskers represent outliers.

**Figure 7 jof-07-00087-f007:**
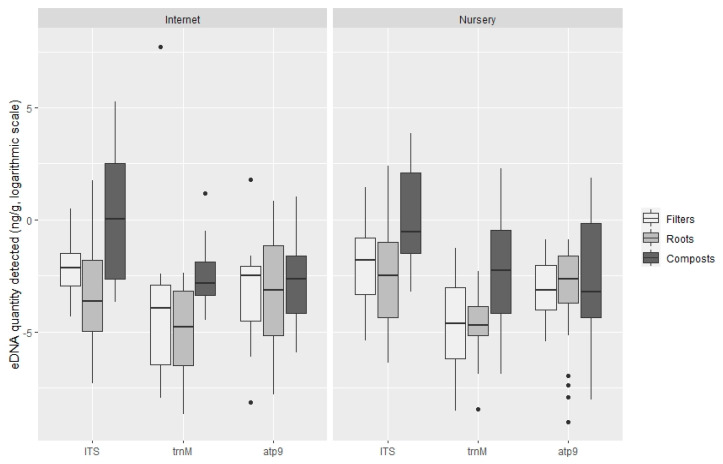
Quantification of oomycete eDNA (ng g^−1^, logarithmic scale) by TaqMan probe in internet sales and plants obtained in nurseries. Boxplots indicate the 25th percentile (Q1, lower boundary), median (black line) and 75th percentile (Q3, upper boundary). Upper and lower whiskers indicate 10th and 90th percentiles. Dots above and below the whiskers represent outliers.

**Table 1 jof-07-00087-t001:** TaqMan primers and probes used in eDNA quantification.

Locus	Probe/Primers	Fluorophore/Quencher of the Probe	Sequence (5′ to 3′)	Annealing Temperature (°C)	Ref
ITS	All-phy Probe	FAM/BHQ1	TTGCTATCTAGTTAAAAGCA	60	[32]
	FITS_15Ph		TGCGGAAAGGATCATTACCACACC		
	RITS_279Ph		GCGAGCCTAGACATCCACTG		
trnM-trnP-trnM	TrnM_PhyG_probe2	FAM/BHQ1	ATRTTGTAGGTTCAARTCCTAYCATCAT	62	[21]
	PhyG-F2		CGTGGG AATCATAATCCT		
	PhyG-Rb		CAGATTATGAGCCTGATAAG		
atp9-nad9	ATP9_PhyG2_probeR	FAM/BHQ1	AAAGCCATCATTAAACARAATAAAGC	57	[21]
	PhyG_ATP9_2FTail		AATAAATCATAACCTTCTTTACAACAAGAATTAATG		
	PhyG-R6_Tail		AATAAATCATAAATACATAATTCATTTTTATA		
PLRV (Internal Positive Control)	PLRV-P-HEX	HEX/BHQ1	CGAAGACGCAGAAGAGGAGCCAAT	60	[33]
	IntConF (PLRV-F)		AAGAGGCGAAGAAGGCAATCC		
	IntConR (PLRV-R)		GCACTGATCCTCAGAAGAATCG		

**Table 2 jof-07-00087-t002:** Plant species analysed, with oomycetes isolated using classical techniques.

Nursery and Country of Purchase	Plant Species	Plant Codes	Oomycetes Species Isolated		
			*Phytophthora* spp.	*Pythium* spp.	*Phytopythium* spp.
Nursery 1—UK (I)	*Viburnum* x *bodnantense* ‘Dawn’	P1	*P. ramorum* (R, S), *P. cryptogea* (B, R)	*P. dissotocum* (B, R), *P. lutarium/diclinum* (B)	*P. litorale* (B, S)
	*Acer palmatum* ‘Dissectum Filigree’	P21, P22, P23, P24	*P. cambivora* (B, S), *P. plurivora* (B)	*P. dissotocum* (B, R, S), *P. lutarium* (B), *P. dissimile/pyrilobum* (R), *P. intermedium* (S), *P. perplexum* (R)	NI
	*Viburnum plicatum* ‘Lanarth’	P25, P26, P27, P28, P29	NI	*P. dissotocum* (B, R), *P. intermedium* (S), *P. deliense* (R), *P. sterilum* (B), *P. kashmirense* (B)	*P. helicoides* (S)
	*Viburnum burkwoodii* ‘Park Farm hybrid’	P30, P31, P32, P33, P34, P45, P46, P47, P48, P49	*P. cryptogea* (B, R, S), *P. chlamydospora* (B, R, S), *P. cactorum* (S)	*P. dissotocum* (B, R), *P. anandrum* (S), *P. dissotocum/coloratum* (B)	*P. litorale* (B, S)
	*Viburnum tinus* ‘Eve Price’	P35, P36, P37, P38, P39	*P. citrophthora* (B), *P. cactorum* (B, S), *P. cryptogea* (B)	*P. dissotocum* (B)	*P. vexans* (S), *P. helicoides* (R), *P. chamaehyphon* (S)
	*Ceanothus thyrsiflorus* ‘Southmead’	P99	*P. cinnamomi* (B), *P. cambivora* (B)	*P. intermedium* (R), *P. dissotocum/diclinum* (B)	NI
Nursery 2—UK	*Acer palmatum* ‘Orange Dream’	P2, P3, P4	*P. plurivora* (B, S)	*P. irregulare* (B), *P. debaryanum/violae* (B)	*P. chamaehyphon* (S), *P. citrinum* (S)
	*Acer palmatum* ‘Atropurpureum’	P5, P6, P7	NI	*P. torulosum/catenulatum* (B), *P. undulatum* (B), *P. intermedium* (B)	NI
	*Hebe* x *franciscana* ‘Variegata’	*P8, *P9, *P10, *P11, *P12, *P13	NI	*P. dissotocum* (B, R, S), *P. diclinum/lutarium* (B)	*P. chamaehyphon* (S), *P. litorale* (S), *P. vexans* (S)
Nursery 3—UK	*Rhododendron* ‘Germania’	*P14, *P15	*P. cinnamomi* (B, R, S)	*P. undulatum* (B, S)	NI
	*Rhododendron* ‘Marcel Menard’	*P16, *P17	NI	*P. undulatum* (B, S), *P. macrosporum* (B, S)	*P. helicoides* (B)
	*Rhododendron* ‘Percy Wiseman’	*P18, *P19	NI	*P. undulatum* (B, S)	NI
Nursery 4—UK (I)	*Camellia alba* ‘Plena’	*P40, *P41, *P42, *P43, *P44	NI	*P. heterothallicum* (R), *P. irregulare* (B), *P. intermedium* (R)	NI
Nursery 5—NL (I)	*Ceanothus thyrsiflorus* ‘Repens’	*P50, *P51, *P52, *P53, *P54	*P. nicotianae* (B), *P. cactorum* (B, R, S), *P. citrophthora* (B, R, S), *P. cinnamomi* (B, S)	*P. sylvaticum* (R, S), *P. sylvaticum/terrestris* (S), *P. dissotocum* (B), *P. lutarium/diclinum* (B), *P. rostratifingens* (B), *P. adhaerens/* *chondricola* (B)	NI
	*Euonymus fortunei* ‘Emerald Gaiety’	P55, P56, P57, P58, P59	NI	*P. dissotocum* (B), *P. rostratifingens* like (R)	*P. vexans* (B)
Nursery 6—NL	*Ilex meserveae* ‘Blue Maid’	P60, P61, P62, P63	*P. plurivora* (B), *P. cinnamomi* (B, R, S), *P. cambivora* (B), *P. cryptogea* (B, S)	*P. dissotocum* (B)	*P. litorale* (B, S), *P. citrinum* (B)
	*Ilex aquifolium* ‘Argentea’	*P64, P65, P66,* P67	*P. chlamydospora* (R)	*P. anandrum* (R, S), *P. dissotocum* (B)	*P. litorale* (B, R)
	*Ilex* x *altaclerensis* ‘Golden King’	*P68, P69, P70, P71	*P. cambivora* (B)	NI	*P. litorale/sterilum* (B, R, S)
	*Pinus mugo*	*P72, *P73, *P74, *P75, *P76, *P77	*P. cambivora* (S)	*P. dissotocum* (R), *P. litorale/sterilum* (R)	*P. litorale* (B)
Nursery 7—NL	*Camellia japonica*	P78, P79, P80, P81	NI	*P. dissotocum* (B), *P. lutarium/diclinum* (B), *P. anandrum* (S)	*P. litorale* (B, S)
	*Buxus sempervirens*	P82, P83, P84, P85, P86	*P. multivora* (B)	*P. sylvaticum* (B), *P. ultimum* var. *ultimum* (S), *P. dissotocum* (B), *P. dissotocum/lutarium* (B), *P. rostratifingens* (R), *P. rostratifingens/camurandrum* (R)	*P. litorale* (B)
	*Rhododendron obtusum* ‘Anouk’	P87, P88, P89, P90	NI	*P. undulatum/ultimum* (S)	*P. litorale* (B, S)
Nursery 8—NL	*Euonymus fortunei* ‘Arlequin’	*P91, *P92, *P93, *P94, P95, P96, P97, P98	*P. multivora* (B), *P. citrophthora/colocasiae* (B), *P. citrophthora* (B), *P. plurivora* (B)	*P. dissotocum/lutarium* (B, R), *P. undulatum/ultimum* (B)	*P. litorale* (B), *P. citrinum/helicoides* (B), *P. citrinum/chamaehyphon* (B), *P. vexans* (S)

I: Internet purchases; *: asymptomatic plant; NI: no species isolated; B: species isolated from baiting, R: species isolated from roots, S: species isolated from substrate.

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
