# Peer review of "Application of Real-Time PCR for the Detection and Quantification of Oomycetes in Ornamental Nursery Stock"

_jof, 2021, doi:10.3390/jof7020087_

Round 1
Reviewer 1 Report
The manuscript entitled: "Application of Real-time PCR for the Detection and Quantification of Oomycetes in Ornamental Nursery Stock", presented for review, shows the results of research on the occurrence of Oomycetes - important pathogens of many plants by use of combination classical and molecular methods. The material for the study was Ninety nine woody ornamental plants, including 23 species imported into Europe.
The design of the manuscript is well thought-out. The research was done and described with great care. As a result of the research it was found that together with the plant material Oomycetes can be easily transferred to different countries. In addition, the qPCR technique used for testing nursery material for Oomycetes was shown to be highly effective. I believe that the manuscript fully deserves to be published in the Journal of Fungi (MDPI) after some really minor changes.
Notes:
Table 1 requires formatting. What does the Modifications column refer to? Primers and/or Probes? aT requires clarification.
L.296. R software vs. 3.5.0?
L.317. Fig. 1 not included in full version. Please correct it.
L.346. Fig. 2 a,b,c, and Fig. 3 are not attached correctly. This needs to be corrected.
L.414. Fig. 4 needs to be corrected.
L417. The signature of Table 3 is mistaken. The table does not show any correlation.
Author Response
The manuscript entitled: "Application of Real-time PCR for the Detection and Quantification of Oomycetes in Ornamental Nursery Stock", presented for review, shows the results of research on the occurrence of Oomycetes - important pathogens of many plants by use of combination classical and molecular methods. The material for the study was Ninety nine woody ornamental plants, including 23 species imported into Europe.
The design of the manuscript is well thought-out. The research was done and described with great care. As a result of the research it was found that together with the plant material Oomycetes can be easily transferred to different countries. In addition, the qPCR technique used for testing nursery material for Oomycetes was shown to be highly effective. I believe that the manuscript fully deserves to be published in the Journal of Fungi (MDPI) after some really minor changes.
We thank the reviewer for these positive comments.
Notes:
Table 1 requires formatting. What does the Modifications column refer to? Primers and/or Probes? aT requires clarification.
Table 1 was re-formatted as suggested by all reviewers. Column titles were revised for clarification.
L.296. R software vs. 3.5.0?
R version number is now specified.
L.317. Fig. 1 not included in full version. Please correct it.
The shape and fit of Figure 1 was corrected in order to be fully displayed
L.346. Fig. 2 a,b,c, and Fig. 3 are not attached correctly. This needs to be corrected.
Figure formatting was corrected.
L.414. Fig. 4 needs to be corrected.
Figure formatting was corrected.
L417. The signature of Table 3 is mistaken. The table does not show any correlation.
Legend of Table 3 modified to remove the word “correlation”. [Table 3 was removed from the main text and transferred to supplementary information.]
Reviewer 2 Report
The article titled ‘application of real-time PCR for the detection and quantification of oomycetes in ornamental nursery stock’ is well written and well organized. The research would help us to know the pathogen, especially oomycetes transfers internationally by ornamental plants. However, the manuscript should be improved by adding or explaining the followings comments-
- L149: explain the media components of CMA-P5ARBP/H
- L229: Normally, we follow ‘Company name, City and Country name’ (Eg. Applied Biosystems, MI, USA)
- Table 1 must be reshaped, and a footnote is needed.
- L302-3-4: If there is no available information about the plants' origin, authors could mention how they got the plants or from which country they collected them. Alternatively, the authors could delete this part.
- Table 2: Caption and footnote should be separated. Write the detailed captions and separate footnotes from the captions.
- Table 3: Not necessary to include in the main manuscript. Use this table as a supplementary table.
- Figure 2,3 & 5 is not complete. Part of the figure is absent in the manuscript. Resize the figure and make it visible. Captions should be more informative.
- The last part of the discussion can be written as a conclusion.
Author Response
The article titled ‘application of real-time PCR for the detection and quantification of oomycetes in ornamental nursery stock’ is well written and well organized. The research would help us to know the pathogen, especially oomycetes transfers internationally by ornamental plants. However, the manuscript should be improved by adding or explaining the followings comments-
- L149: explain the media components of CMA-P5ARBP/H
The medium components used here were precisely the same as those described by Jeffers & Martin (1986): we decided to include the reference to the culture medium instead of the full recipe. The comment of Reviewer 2 about line 149 is resolved here (reference [23]).
L229: Normally, we follow ‘Company name, City and Country name’ (Eg. Applied Biosystems, MI, USA)
All company names have been modified, as suggested.
Table 1 must be reshaped, and a footnote is needed.
Table 1 was revised, as suggested.
L302-3-4: If there is no available information about the plants' origin, authors could mention how they got the plants or from which country they collected them. Alternatively, the authors could delete this part.
Country in which the plants were sourced is now added.
Table 2: Caption and footnote should be separated. Write the detailed captions and separate footnotes from the captions.
Information was moved from the caption to a footnote under Table 2.
Table 3: Not necessary to include in the main manuscript. Use this table as a supplementary table.
Table 3 was transferred to the supplementary material, as suggested.
Figure 2,3 & 5 is not complete. Part of the figure is absent in the manuscript. Resize the figure and make it visible. Captions should be more informative.
The figures were fully visible at the time of submission, but the margins changed online. We re-sized the figures to bring every part into view.
The last part of the discussion can be written as a conclusion.
The last paragraph was sub-titled ‘conclusions’ and the text modified slightly.
Reviewer 3 Report
This study was aimed at verifying if potted ornamental plants purchased through internet sales pose a risk of dissemination of oomycete plant pathogens. The presence of Phytophthora, Pythium and Phytopythium species in compost, roots and water extract from compost was determined using conventional leaf baitings and successive isolation on selective medium while the total amount of oomycetes and Phytophthora was quantitatively determined by Real-time PCR using three targeted loci ITS, trnM-trnP-trnM and 22 atp9-nad9. No significant differences were found in quantities of oomycete DNA detected using real-time PCR in plants purchased online or from traditional retailers. These results draw attention to the risk of dissemination of plant pathogens through the internet trade.
I would have expected the Authors to propose solution(s) to the problem in the Discussion and to cite the promising preliminary results obtained by applying molecular methods to the diagnosis of species of Phytophthora pathogenic for potted ornamental plants also by including previous published studies.
The numbering of References is missing.
For other minor comments see the notes in the text (attached file)

Author Response
This study was aimed at verifying if potted ornamental plants purchased through internet sales pose a risk of dissemination of oomycete plant pathogens. The presence of Phytophthora, Pythium and Phytopythium species in compost, roots and water extract from compost was determined using conventional leaf baitings and successive isolation on selective medium while the total amount of oomycetes and Phytophthora was quantitatively determined by Real-time PCR using three targeted loci ITS, trnM-trnP-trnM and 22 atp9-nad9. No significant differences were found in quantities of oomycete DNA detected using real-time PCR in plants purchased online or from traditional retailers. These results draw attention to the risk of dissemination of plant pathogens through the internet trade.
I would have expected the Authors to propose solution(s) to the problem in the Discussion and to cite the promising preliminary results obtained by applying molecular methods to the diagnosis of species of Phytophthora pathogenic for potted ornamental plants also by including previous published studies.
The numbering of References is missing.
The references were numbered in the version uploaded, but the system seems to have deleted them – we have corrected the problem.
For other minor comments see the notes in the text (attached file)
Lines 28-29: meaning of the data was clarified.
Line 131: The apple variety ‘Granny Smith’ is a trademark, so we added ™ after the name, rather than the copyright symbol (©) as suggested by the reviewer.
Table 1: edited, as suggested.
Line 541: the papers proposed have been included in the discussion section
Line 575: As with formatting of the figures, the references were numbered in the original manuscript, before it was uploaded.
Line 636: references added here (and at appropriate points in the text).
Round 2
Reviewer 2 Report
Dear Author,
All of you have done a very good job to modify the article.
But still the figure 1 is not fully visible.
Others are scientifically sound.